



# Fine-scale variability in iceberg velocity fields and implications for an ice-associated pinniped

Lynn M. Kaluzienski[1], Jason M. Amundson[1], Jamie N. Womble[2], Andrew K. Bliss[2], and Linnea E. Pearson[2]

[1]Department of Natural Sciences, University of Alaska Southeast, 11066 Auke Lake Way, Juneau, 99801, Alaska, USA
[2]Southeast Alaska Network and Glacier Bay National Park and Preserve, National Park Service, 3100 National Park Road, Juneau, 99801, Alaska, USA

**Correspondence:** Lynn M. Kaluzienski (lmkaluzienski@alaska.edu)

**Abstract.** Icebergs and sea ice found in proglacial fjords serve as important habitat for pinnipeds in polar and subpolar regions. Environmental forcings can drive dramatic changes in fjord ice coverage, with implications for pinniped distribution, abundance, and behavior. To better understand how pinnipeds respond to changes in iceberg habitat, we combine (i) iceberg velocity fields over hourly to monthly timescales, derived from high-rate time-lapse photogrammetry of Johns Hopkins Glacier and Inlet, Alaska, with (ii) aerial photographic surveys of harbor seals (*Phoca vitulina richardii*) conducted during the pupping (June) and molting (August) seasons. Iceberg velocities typically followed a similar diurnal pattern: flow was weak and variable in the morning and strong and unidirectional in the afternoon. The velocity fields tended to be highly variable in the inner fjord, across a range of timescales, due to changes in the strength and location of the subglacial outflow plume, whereas in the outer fjord the flow was more uniform and eddies consistently formed in the same locations. During the pupping season, seals were generally more dispersed across the slow moving portions of the fjord (with iceberg speeds <0.2 m s$^{-1}$). In contrast, during the molting season the seals were increasingly likely to be found on fast moving icebergs in or adjacent to the glacier outflow plume. Use of slow moving icebergs during the pupping season likely provides a more stable ice platform for nursing, caring for young, and avoiding predators. Periods of strong glacier runoff and/or katabatic winds may result in more dynamic and less stable ice habitat, with implications for seal behavior and distribution within the fjord.

## 1 Introduction

Ice, including sea ice and icebergs, provides important habitat for marine mammals in subpolar and polar regions (Kelly, 2001; Laidre et al., 2015). For pinnipeds, ice provides a substrate for pupping, nursing young, avoiding predators, and reducing the likelihood of disease transmission (Fay, 1974). Given the dramatic changes in ice coverage and the reliance of pinnipeds on ice habitat for critical life-history functions (e.g, Fay, 1974; Kelly, 2001; Laidre et al., 2015; Gulland et al., 2022), an understanding





of ice dynamics and variability across multiple spatial and temporal scales is essential for projecting how changes in climate may influence the distribution, abundance, and behavior of pinnipeds.

Tidewater glacier fjords in Alaska host some of the largest seasonal aggregations of harbor seals in the world (Jansen et al., 2015). In Glacier Bay (Sít' Eetí G̲eeyi), harbor seals are monitored using aerial photographic surveys approximately 6–8 times per year during the pupping (June) and molting (August) seasons in order to quantify their abundance and spatial distribution (Womble et al., 2020, 2021). The surveys also provide important information regarding the ice habitat used by seals (McNabb et al., 2016; Womble et al., 2021; Kaluzienski et al., 2023); however, the aerial photographic methods were designed to estimate the abundance of seals over broad temporal and spatial scales (Ver Hoef and Jansen, 2015). While it is well-documented that tidewater glacier fjords provide important habitat for harbor seals, we lack an understanding of the physical processes and environmental factors that influence the fine-scale variability of iceberg habitat in the fjord and how this variability influences the distribution and behavior of seals.

Recent advances in camera and computer technologies have enabled the development of tools for tracking fine-scale spatial and temporal variability in iceberg habitat in tidewater glacier fjords with time-lapse photography (Kienholz et al., 2019) over time periods of minutes to years. Here, we use the methods of Kienholz et al. (2019) in order to characterize iceberg velocity fields during summer in Johns Hopkins Inlet (Tsalx̲aan Niyaadé Wool'éex'i Yé), which is located in the northwest corner of Glacier Bay (Fig. 1), and relate the velocity fields to the abundance and distribution of seals within the fjord.

## 2 Study area and methods

Johns Hopkins Inlet (Fig. 1) is a tidewater glacier fjord in Glacier Bay National Park and Preserve in southeastern Alaska. We focus on the portion of the fjord that extends from Johns Hopkins Glacier to Jaw Point, which is about 9 km long and 1.5–2 km wide. Johns Hopkins Glacier, the primary tidewater glacier feeding into the fjord, reached a minimum extent in the 1920s following the disintegration of the Glacier Bay Icefield (Hall et al., 1995); it has since advanced 2 km and thickened by over 100 m in its lower reaches (Larsen et al., 2007; McNabb and Hock, 2014). In recent years shoaling of the glacier's end moraine has reduced the iceberg flux into the fjord, thereby reducing ice habitat for seals (Kaluzienski et al., 2023). Johns Hopkins Inlet is also fed by Gilman Glacier, whose terminus coalesces with that of Johns Hopkins Glacier. Gilman Glacier is an order of magnitude smaller than Johns Hopkins Glacier by area and is more stable, having advanced by about 200 m over the last century (McNabb and Hock, 2014). Thus, the impact of Gilman Glacier on iceberg production and habitat is relatively small compared to Johns Hopkins Glacier.

### 2.1 Time-lapse cameras

We deployed four time-lapse cameras in Johns Hopkins Inlet during the summers of 2019, 2021, and 2022. Due to camera malfunctions and a lack of overlapping aerial surveys in 2021, we focus exclusively on the data that was collected in 2019 and 2022. The time-lapse camera systems consisted of 18 mm Canon Rebel T3 or T5 single-lens reflex (SLR) cameras that were controlled by Harbortronics DigiSnap intervalometers. Two cameras were co-located in the inner fjord (<3 km from the



**Figure 1.** Map of the study site showing (a) Johns Hopkins Inlet and Glacier Bay and their location within (b) Alaska and (c) Glacier Bay National Park and Preserve. In (a), small white boxes indicate image footprints from an aerial survey flown on 29 July 2019 and colored lines denote each time-lapse camera's field of view. In (c), green star indicates Tarr Inlet tidal station and yellow and blue triangles indicate Queen Inlet and Lone Island weather stations, respectively. (d) Example photos from camera 1 taken in 2019. (e) Terminus positions for Johns Hopkins and Gilman Glaciers during summers 2019–2023. The background images in (a) and (e) are Sentinel-2 images from 2023 with Alaska Albers Projection (EPSG:3338).




glacier) and two cameras were co-located at Jaw Point (∼8.5 km from the glacier, Fig. 1). We changed the location of the cameras in the inner fjord after the 2019 season to a site that is more accessible and that has a better view of the glacier. The

cameras were installed on survey grade tripods that were stabilized with large piles of rocks, oriented to collectively observe the entire fjord and have some overlap across photos, surveyed with geodetic quality GNSS receivers (Emlid Reach RS2+), and programmed to take photos every minute for 14–16 hrs each day from June to September. One camera system malfunctioned each season due to either disturbance from terrestrial wildlife (e.g., bears) or electrical issues.

We calculated iceberg velocities from the time-lapse photos using the workflow described in detail in Kienholz et al. (2019),

which involved building camera models, tracking features, and using the camera model to translate pixel displacements into map coordinates. We used a simple, planar camera model that has four free parameters: yaw, pitch, roll, and focal length (Krimmel and Rasmussen, 1986). We determined these parameters by digitizing the fjord waterline in a representative photo, using an initial guess to project the waterline into map view coordinates, and iteratively adjusting the parameters in order to minimize the distance between the projected waterline and the waterline observed in a coincident Landsat 8 image. The process

required knowledge of the camera's location, which was known from the GNSS surveys, and elevation relative to sea level. The GNSS solutions provided the camera's ellipsoidal elevation. We determined the ellipsoidal sea level elevation by shifting the NOAA tide prediction curve for nearby Tarr Inlet (https://tidesandcurrents.noaa.gov/noaatidepredictions.html?id=9452749) so that it agreed with the local ellipsoidal sea level elevations from IceSAT-2 ATL06 data (Smith et al., 2023). Comparison between the tide prediction and previous in situ tide measurements showed good agreement with the timing and magnitude of

the tides in Johns Hopkins Inlet.

Once the camera models were determined, we applied a highpass filter to the images, identified features with the Shi-Tomasi algorithm (Shi and Tomasi, 1994), and tracked the features with the Lucas-Kanade sparse optical flow algorithm (Lucas and Kanade, 1981) as implemented in OpenCV (https://opencv.org). To filter erroneous calculations, we tracked features over four successive images and excluded any features that could not be tracked over all four images. The pixel displacements were then

converted to map view; changes in tidal elevation were accounted for during this conversion. The resulting sparse velocity field was gridded and used to create streamline plots and plot velocity transects. Kienholz et al. (2019) demonstrated that this workflow can produce velocities with errors of less than 0.1 m s$^{-1}$ and that the error becomes smaller when the velocity fields are temporally averaged. In addition, they also demonstrated that small icebergs, such as those found in Johns Hopkins Inlet, serve as good tracers of surface water currents.

**2.2  Aerial photographic surveys**

Six to eight aerial photographic surveys have been carried out in Johns Hopkins Inlet most summers since 2007 in order to monitor harbor seals during the pupping (June) and molting (August) periods. Until 2019 the surveys followed the methodology outlined in Womble et al. (2020, 2021) and summarized below. No surveys were conducted in 2020 and challenges associated with transitioning to a new camera system in 2021 precluded collection of nadir photos that could be georeferenced; as stated

above, we therefore focus exclusively on data collected in 2019 and 2022.

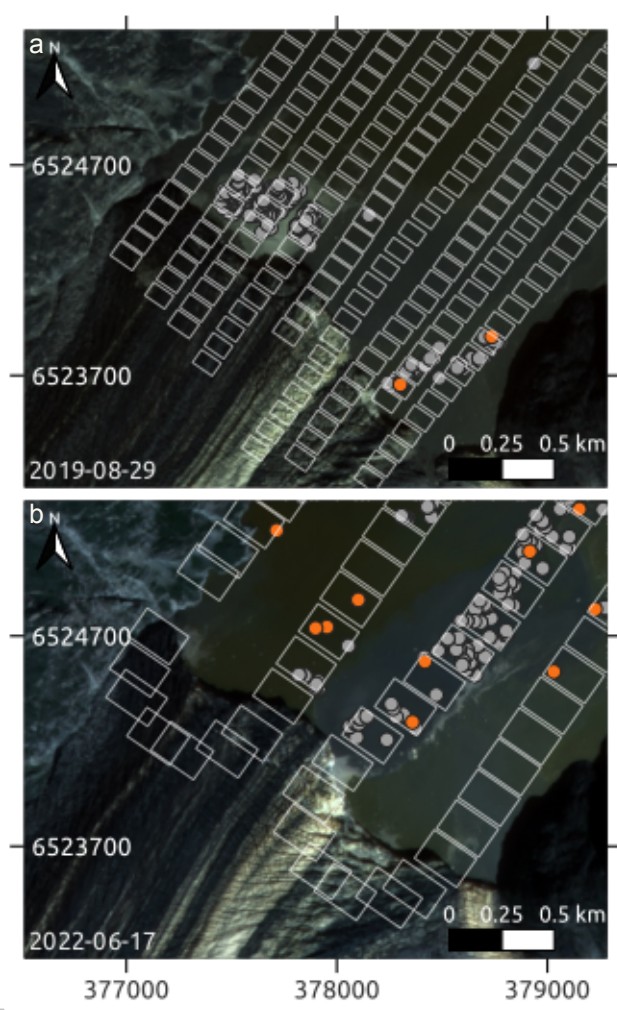

**Figure 2.** Footprints of aerial photographs in the terminus region during the 2019 and 2022 surveys. Orange and gray dots represent seals identified in the water and hauled out on icebergs, respectively. Background images were taken by the Planet satellite on (a) 29 August 2019 and (b) 21 August 2022 and projected into WGS84 UTM Zone 8N (EPSG:32608). Images ©2019 and 2022 Planet Labs.



In 2019 the surveys were conducted from a de Havilland Canada DHC-2 Beaver. The aircraft flew transects at an altitude of ~304 m and speed of 166 –175 km/h; the transects were aligned with the fjord and spaced ~200 m apart. A digital, single-lens reflex camera (Nikon D2X, 12.4 megapixel) was mounted to a platform and secured to the belly porthole of the aircraft. Photographs were taken every 2 s with a digital timer (Nikon MC36) and the trackline and position of the aircraft was recorded at the same interval with an onboard global positioning system (Garmin 76 CSX). The camera interval, aircraft speed, and transect spacing ensured that the photographs did not overlap and therefore that seals would not be counted twice. There was a roughly 15 m gap between successive photographs and 70 m gap between photographs on adjacent transects. Each photograph covered an area of approximately 80 m×120 m with a resolution of ~3.24 cm/pixel.

In 2022 a new camera system was used and surveys were conducted using a lightweight camera pod (Waldo XCAM Ultra 50) that was attached to the wing strut of a Cessna 206 aircraft. The aircraft flew at an altitude of ~293 m and speed of 148 – 168 km/h. The camera pod consisted of two Canon 5DS R 50.6 MP RGB cameras with fixed-zoom 50mm f/1.4 lens, an integrated GPS unit, and a micro-controller to trigger capture events. The two cameras in the pod were synchronized by Waldo software and programmed to capture images every 2 s; the aircraft location was also recorded every 2 s. The footprint of each photograph was 159 m×470 m, resulting in a resolution of ~2.75 cm/pixel. Due to the larger footprint of these new system, only four transects were flown to cover an area similar in size to previous surveys, with an approximate distance of ~210 m between transects. While the area coverage is similar between years, the 2022 data has larger horizontal gaps between photographs (Fig. 2).

For all surveys, the photographs were georeferenced using the aircraft latitude, longitude, and altitude and the camera resolution and focal length. The georeferenced images were then inspected to identify seals and create shapefiles containing the seal locations. Shape files for both pups and nonpups were created during the pupping season, while only shape files for nonpups were created during the molting season.

The aerial surveys were carried out under NOAA Fisheries Marine Mammal Protection Act (MMPA) permit numbers 358-1787-00, 358-1787-01, 358-1787-02, and 16094-02. For more information on the survey methods, refer to Womble et al. (2020).

## 2.3 Relating seal distributions to fjord conditions

The speed that an iceberg is traveling that a seal is hauled out on depends on both the background flow speed of the fjord and the iceberg's location within the fjord. In order to quantify the likelihood of a seal being found on an iceberg or on the surface of relatively fast or slow moving water, we therefore compare the velocity distribution of the seals to the background velocity distribution of the fjord. We first calculated the average iceberg velocity within each 50 m × 50 m grid cell during the three-hour window of each aerial survey ("fjord velocity"), and then associated each seal with the velocity of the grid cell that it was located within ("seal velocity"). We next averaged the fjord velocity across each of the surveys during the 2019 molting and 2022 pupping seasons, respectively. These data were then used to compute complementary cumulative distribution functions (CCDFs), which indicate the likelihood that speed $V$ of a randomly selected grid cell or seal will have a speed that is greater than $v$.





| Date | Planet Imagery | Cloud Cover | Temp (°C) | Mean Speed (m/s) | Seals in Water | Seals on Ice |
|---|---|---|---|---|---|---|
| 2019-8-12 | yes | clear | 13 | 0.12 +/- 0.08 | 22 | 473 |
| 2019-8-16 | yes | high overcast then clear | 13 | 0.13 +/- 0.09 | 11 | 687 |
| 2019-8-29 | yes | clear | 14 | 0.12 +/- 0.08 | 14 | 649 |
| 2019-8-30 | yes | clear | 16 | 0.11 +/- 0.11 | 61 | 224 |
| 2022-6-16 | yes | clear | 20 | 0.11 +/- 0.07 | 135 | 489 |
| 2022-6-17 | no | clear | 15 | 0.09 +/- 0.07 | 52 | 962 |
| 2022-7-05 | no | high overcast | 19 | 0.13 +/- 0.09 | 44 | 271 |

**Table 1.** Data availability, cloud cover, air temperature, daily average speed of fjord, and number of seals in water and on ice for relevant aerial photographic surveys in June, July, and August of 2019 and 2022. Cloud cover and air temperature were recorded during aerial surveys.

## 2.4 Auxiliary and Meteorological data

To better understand fjord conditions during our aerial photographic surveys, we supplemented aerial surveys with 3-m resolution optical satellite imagery from PlanetScope, obtained through NASA's Commercial SmallSat Data Acquisition Program (Planet Team, 2017) when available (Fig. 1). In particular, we manually digitized outlines of glacial sediment plumes and regions of recent calving events to compare with velocity patterns and seal locations within the fjord.

To help interpret the variability in iceberg habitat that we observed, we also analyzed temperature and precipitation data at Lone Island and Queen Inlet (Fig. 1), which are 47 km and 25 km away from Johns Hopkins Inlet, respectively, and at elevations of 26 m and 319 m above sea level. Both sites consist of Campbell Scientific research grade weather stations and are part of a long-term monitoring network to record weather and climate conditions in Glacier Bay and other coastal parks in southeastern Alaska (Bower et al., 2017). Temperature was measured within a naturally-aspirated radiation shield (sensor tolerance of 0.2°C) every 60 s and averaged hourly; precipitation was measured continuously with a tipping bucket (resolution of 0.0254 cm) and logged hourly. Due to instrument malfunction, we do not use the precipitation data from Lone Island and, although the stations also recorded other meteorological parameters, such as wind speed and direction, we exclude them from our analysis due to the effects of local topography on winds. The agreement between the temperature recorded by the two weather stations (shown in Fig. 4) indicates that they are sufficient for quantifying synoptic scale climate variability within Glacier Bay.

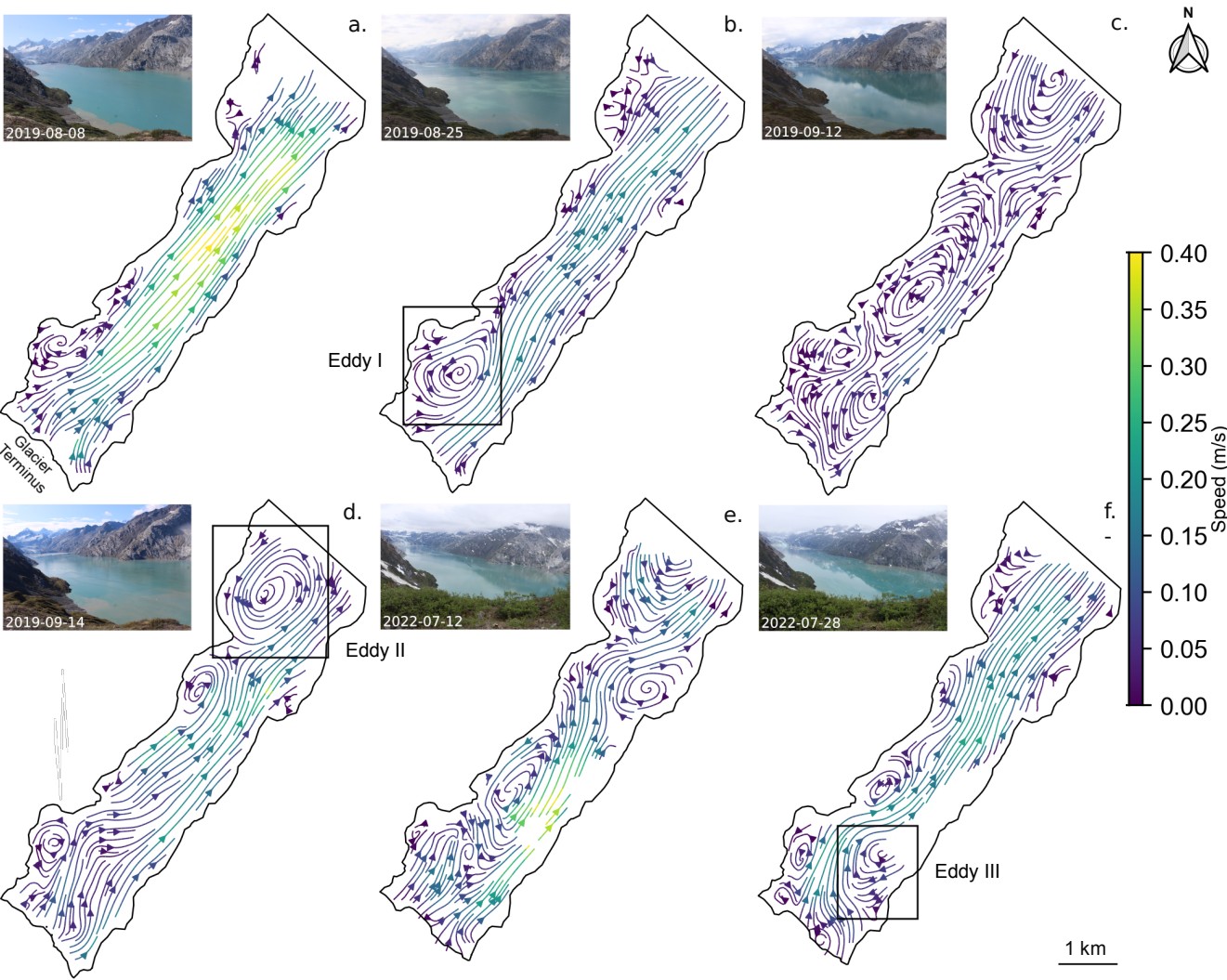

**Figure 3.** Streamlines derived from daily average velocity fields for select days throughout the 2019 and 2022 field seasons. Inset photos taken from camera 3 show fjord conditions for each day. Prominent, persistent eddies are indicated in (b), (d), and (f). All maps are in WGS84 UTM Zone 8N (EPSG:32608).



## 3 Results

The iceberg tracking algorithm produced spatially complete daily-averaged velocity fields (Fig. 3) during approximately 80% of the study period, which spanned from 20 July–17 September 2019 and 15 June–26 August 2022. The fjord-averaged velocity was always in the down fjord direction, with speeds typically ranging from 0.05–0.20 m s$^{-1}$ with a standard deviation of $\sim$0.1 m s$^{-1}$ (Fig. 4). Thus, iceberg residence times within the study area from Johns Hopkins Glacier to Jaw Point, a distance of 9 km, ranged from 0.5–2 d. Iceberg speed distributions were similar in June and August (average speed : 0.10 m s$^{-1}$), but were faster in July (average speed : 0.12 m s$^{-1}$).

Large eddies commonly formed in the fjord, and frequently formed at the three locations labeled I, II, and III in the streamline plots of Figure 3. Eddy III was only observed during 2022. The eddies are also apparent in the time series of transverse velocity profiles of the fjord (Fig. 5). Each column in Figure 5 represents the velocity perpendicular to the given transect on a specific date, with red colors indicating flow away from the glacier and blue colors indicating return flow toward the glacier. Periods with strong eddies therefore appear as vertical slices containing deep red and blue colors. In the inner fjord near the terminus of the glacier, the region of return flow shifted back and forth across the fjord, especially during 2022. In contrast, eddy II was present in the outer fjord on most days. Large eddies were also occasionally observed in the middle of the fjord (Fig. 3c,e and middle panel of Fig. 5).

An active glacial sediment plume was found on all aerial survey days with available optical satellite imagery from PlanetScope (Table 1) and typically extended 4–5 km from the glacier terminus. The outflow of the plume primarily occured on the eastern side (Fig. 8a and c) or central portion of the terminus (Fig. 8b). The plume location closely aligned with the velocity streamline data with regions of faster flow within the plume extent and eddies forming along the periphery. Additionally, icebergs tended to cluster around the edges of the plume.

The velocity fields typically experienced similar diurnal variations (Figs. 4 and 6). During morning hours between 05:00–13:00 (UTC-08:00), when runoff and katabatic (down glacier) winds were likely weak, iceberg velocities were also weak and variable. The velocities typically became stronger and more uniform between 13:00-21:00 (UTC-08:00) as the air temperature rose and runoff and winds increased. Tides do not appear to be a primary driver of flow variability at the fjord surface (Fig. 7). No clear patterns emerged on weekly or longer timescales, suggesting that larger scale circulation within Glacier Bay regulates the base flow upon which the diurnal variations are superposed.

Aerial photographic surveys of seals overlapped with the time-lapse photography campaign across seven dates (Table 1). The relatively small sample size precludes a comprehensive statistical analysis of seal distributions relative to ice velocity fields. We therefore focus on spatial patterns found across the two complimentary data sets. During the the molting period in 2019 (survey dates: 12, 16, 29, and 30 August) the majority of seals were hauled out on icebergs in the inner portions of the fjord near the glacier terminus where icebergs were present from recent calving events. However, on 29 August, the seals were distributed much more extensively along the length of the fjord in addition to two clusters of seals near the termini of the Johns Hopkins and Gilman Glaciers. In contrast, during the pupping period in 2022 (survey dates: 16 and 17 June), seals were more uniformly distributed along the fjord and less clustered near the glacier terminus. On 5 July, the majority of the seals were



**Figure 4.** Meteorological data from stations in Queen Inlet and Lone Island including temperature (a-b) and precipitation (c-d) for 2019 and 2022 field seaons. (e-f) Concurrent hourly mean iceberg speed from iceberg tracking shown in black along with standard deviation in gray.





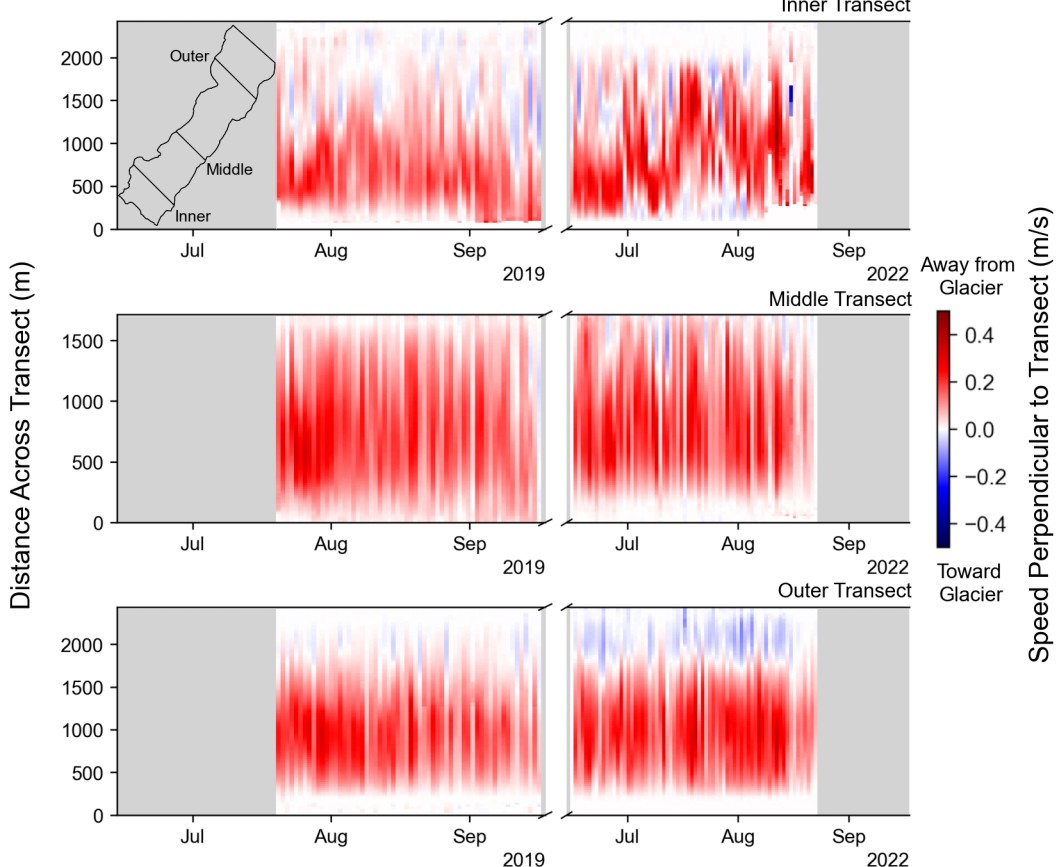

**Figure 5.** Time series of daily average speeds across inner, middle, and outer transects for the 2019 and 2022 field seasons. The inset map shows transect locations within the fjord. Colors represent the speed perpendicular to the transect with red tones indicating positive (down fjord) flow and blues indicating negative (up fjord) flow. The distance across each transect is measured from the eastern side of the fjord.

170    found on icebergs in the outer portion of the fjord near Jaw Point and corresponded to increased iceberg speed. Occasionally seals were observed in the water, traveling or floating at the surface, and not hauled out on icebergs. However, due to glacial silt in water, only seals that were in the upper meter of the water column were available to be detected during surveys.

We observed several patterns when linking seal locations to fjord velocities and plume extent. First, the seals were often located on icebergs within or along the edge of the plume and close to the glacier terminus (Fig. 8a–c), near regions of recently

175    calved icebergs. When seals were observed in water, they were predominantly found within the plume region. Second, when iceberg velocities were high (e.g., Fig. 8d), such as 5 July, the seals were typically found in the outer part of the fjord closer to Jaw Point, several kilometers from the glacier terminus. Finally, in the molting season the seals were more commonly found on relatively fast moving icebergs than during the pupping season.





**Figure 6.** (a-b) Streamlines derived from velocity fields averaged over six hours during the morning (a) and evening (b) of 12 August 2019. (c-e) Time series of hourly average speeds across the inner, middle, and outer transects for the same day. Colors represent the speed perpendicular to the transect with red tones indicating positive (down fjord) flow and blues indicating negative (up fjord) flow. The distance across each transect is measured from the eastern side of the fjord.



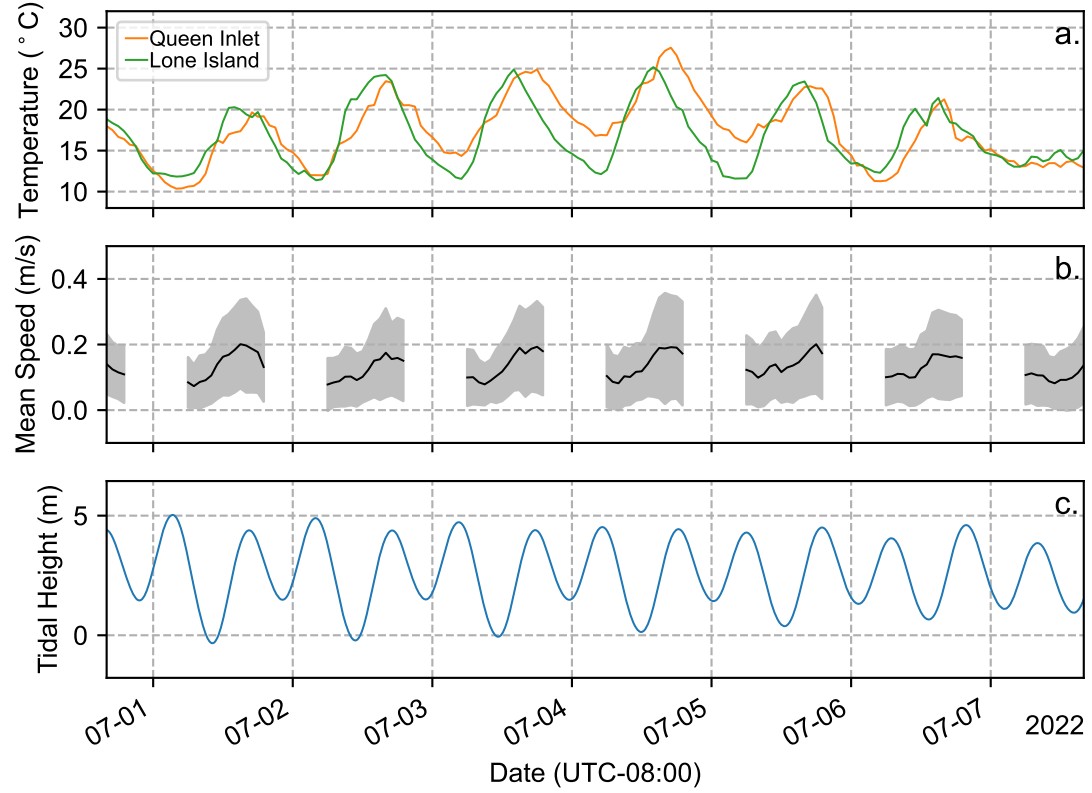

**Figure 7.** (a) Air temperature recorded at the Queen Inlet and Lone Island stations. (b) Mean and standard deviation of the flow speed within the fjord. (c) Predicted tide elevation for the Tarr Inlet tidal station.

The higher likelihood of finding seals on fast moving icebergs on our survey dates from the molting season can be seen by comparing the CCDFs of the fjord and seal velocities (Fig. 9). For both the 2019 molting and 2022 pupping seasons about 20% of the fjord surface was flowing faster than 0.22 m s$^{-1}$. Differences in the CCDFs of the seal velocities from 2019 to 2022 are largely attributable to differences in fjord circulation patterns during these two time periods. However, the CCDFs of the seal velocities do not exactly match those of the fjord velocities. When the CCDFs of the seal velocities lie above the respective curves for the fjord velocities, then there is a higher probability of randomly selecting a seal at that velocity than there is of randomly selecting a fjord cell at that same velocity. The difference between the fjord velocity and seal velocity CCDFs is most apparent during the 2019 molting season. Approximately 15% of the observed seals were located in regions of the fjord that were moving faster than 0.32 m s$^{-1}$, yet only 5% of the fjord was moving at those speeds, suggesting that during the 2019 molting period, seals tended to use icebergs in faster flowing regions of the fjord (i.e., near the plume). In contrast, during the 2022 pupping period, fewer than 5% of the seals were found in regions of the fjord that were flowing faster than 0.25 m s$^{-1}$, which is less than predicted from the fjord velocity distribution.





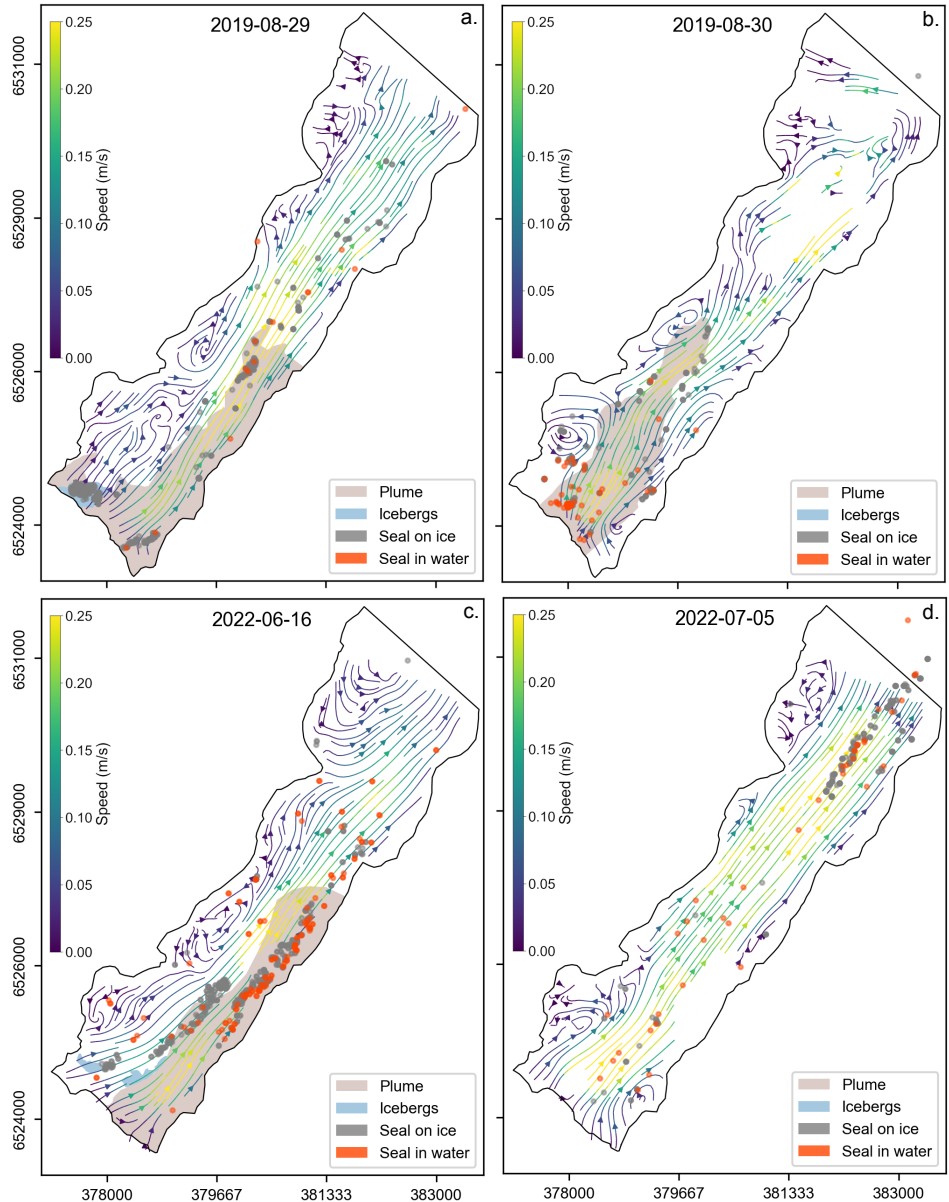

**Figure 8.** Streamlines derived from velocity fields averaged over four-hour periods that coincided with the timing of aerial seal surveys conducted in the molting season in 2019 (a–b) and the pupping season in 2022 (c–d). Time periods were centered on the midpoint of the aerial survey times. Seals spotted on icebergs and in surface waters are marked with gray and orange dots, respectively. Concurrent Planet satellite imagery was used to identify the outlines of plumes and regions with recently calved icebergs, which are indicated in brown and blue, respectively. No plume data was available for 5 July 2022 due to poor satellite image quality.



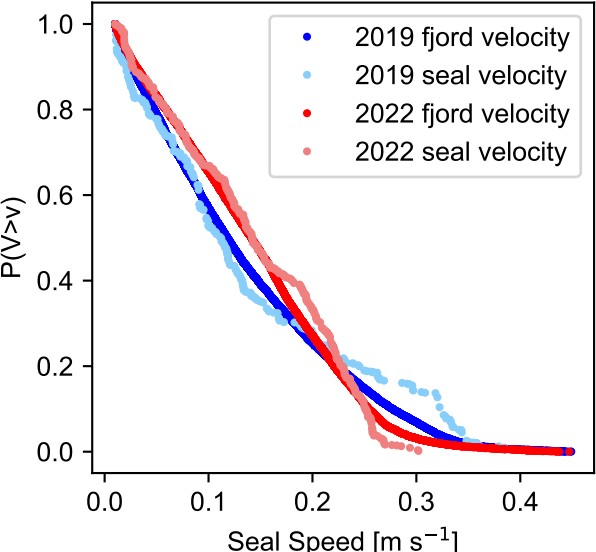

**Figure 9.** Complementary cumulative distribution plot for all aerial survey dates. All surveys in 2019 are from August whereas all surveys in 2022 are from June and early July. We therefore treat 2019 data as representative of the molting season and 2022 as representative of the pupping season.

## 4 Discussion

Freshwater runoff from tidewater glaciers emerges at depth, mixes with warm, salty water, and creates buoyant plumes that rise along glacier termini (Straneo and Cendese, 2015; Truffer and Motyka, 2016). Once a plume reaches neutral buoyancy, which typically occurs at the fjord surface in Alaska during summer (e.g., Walters et al., 1988; Bartholomaus et al., 2013; Jackson et al., 2022), it turns and flows down fjord. The rising plumes carry nutrients and zooplankton that are utilized by seabirds, such as black-legged kittiwakes (*Rissa tridactyla*), and marine mammals (Lydersen et al., 2014; Stempniewicz et al., 2017; Urbanski et al., 2017; Bertrand et al., 2021).

Our observations provide support for the idea that tidewater glacier termini are biological hotspots (Lydersen et al., 2014; Urbanski et al., 2017) and that physical processes occurring at and near the glacier terminus can a play a role in influencing iceberg habitat and the distribution and behavior of harbor seals in the fjord. Whether hauled out on ice or swimming, seals are often found within a few hundred meters of Johns Hopkins Glacier and their location extends throughout the fjord to Jaw Point. Depending upon the availability of iceberg habitat, seals are often either within or immediately adjacent to the outflowing plume (Fig. 8), consistent with observations of ringed seals (*Pusa hispida*) in Svalbard (Everett et al., 2018).



In addition to seals, black-legged kittiwakes from an active colony approximately 0.5 km from the terminus of the Johns
Hopkins Glacier, are regularly observed foraging in the surface waters near the terminus after glacier calving events. Kittiwakes
use the "brown zone", the area around the subglacial discharge that includes large amounts of suspended sediments, to forage
on euphausiids, which has also been documented in numerous other fjords (Urbanski et al., 2017; Stempniewicz et al., 2017).

These data suggest that additional factors, such as velocity fields, iceberg movement, and life-history constraints, may
influence selection of icebergs by seals. During the pupping season, seals were more commonly found on slow moving icebergs,
which are farther from the center of the plume, whereas during the molting season seals were more frequently observed on
fast flowing icebergs ($>0.3$ m s$^{-1}$). When adult females are caring for dependent pups, they may prefer slower and more
stable icebergs as they spend more time hauled out during lactation. In contrast, during the molting period, adult females are
no longer constrained by the presence of a dependent pup and there is presumably less of a need to haul out on ice that is
more stable. Previous studies have also demonstrated the abundance of seals is lower during the molting period than during
the pupping period (Womble et al., 2021) and that fidelity of seals to Johns Hopkins Inlet is reduced during the post-breeding
season (Womble and Gende, 2013).

As demonstrated by Kienholz et al. (2019), the iceberg velocity fields can be treated as estimates of fjord surface currents.
The average currents that we observed (0.05–0.20 m s$^{-1}$), and even the maximum currents (0.4 m s$^{-1}$) are below the measured
minimum cost of transport (MCOT) estimated for swimming juvenile and adult harbor seals, which ranges between between
1.0–1.4 m s$^{-1}$ Davis et al. (1985). Thus, the surface currents in tidewater glacier fjords generally would not energetically
impact swimming adult harbor seals. MCOT has not been measured for dependent pups, which are often observed swimming
alongside their mothers, but it is unlikely that swimming against the average flow rate in Johns Hopkins Inlet would negatively
impact pups.

To further illustrate these findings, we calculate the work required by an adult seal to swim against the current for the full
length of the fjord ($\sim$9 km from Jaw Point to Johns Hopkins Glacier). We approximate the drag force acting on a seal as

$$F_d = \frac{1}{2} C_d \rho_w A v^2, \tag{1}$$

where $C_d$ is the drag coefficient, $\rho_w$ is the density of water, $A$ is the seal's frontal area, and $v$ is the speed of the seal with
respect to the water. Note that the values for $C_d$ and $A$ depend on how the seal area is defined, but the product of $C_d A$ is
unaffected by that definition. The work done by a seal to swim the length of fjord $L$ is $F_d \Delta x$, where $\Delta x$ is the distance that
the seal swims with respect to the water, which is moving with velocity $v_w$:

$$\Delta x = L \left( \frac{v}{v + v_w} \right). \tag{2}$$

Thus the work done is

$$W = \frac{1}{2} C_d \rho_w A v^2 L \left( \frac{v}{v + v_w} \right) \tag{3}$$

Williams and Kooyman (1985) investigated the hydrodynamics of harbor seals and found that $C_d$ is around 0.1, and also report
that the cross-sectional area of harbor seals is about 0.1 m$^2$ and that typical swimming speeds range from about 1.5–2.0 m s$^{-1}$.





Using these values, along with $v_w = -0.4$ m s$^{-1}$ (the negative sign implies that a seal is swimming against the current) yields 33–55 kcal of work to overcome the drag forces acting on the seal (i.e., assuming 100% efficiency). The total cost of transport for a harbor seal swimming at 1.5 m s$^{-1}$ is about 2.5 J m$^{-1}$ kg$^{-1}$ (Davis et al., 1985). Assuming a mass of 85 kg and using the distance in Equation 2 results in a cost of transport of 200–220 kcal, which is a small fraction of the roughly 6000 kcal

that seals consume every day (Härkönen and Heide-Jørgensen, 1991; Rosen and Renouf, 1998). Therefore fjord currents likely have little, direct impact on harbor seals. Currents have many indirect impacts, though, such as (i) affecting the distribution of water masses, nutrients, and small prey, (ii) contributing to melting of glacier termini and icebergs, the former of which affects iceberg calving rates (Ma and Bassis, 2019), and (iii) affecting the distribution of icebergs within a fjord.

## 5  Conclusions

High-rate time-lapse photography reveals that iceberg velocity fields in Johns Hopkins Inlet are highly variable, both temporally and spatially, during summer. During the morning hours, weak flow and large, slow flowing eddies were generally observed. As the day progresses and temperatures and katabatic winds increase, the flow speeds increase and tend to become more uniform with eddies becoming smaller and more icebergs being directed down fjord towards Jaw Point. Eddy locations are variable in the inner fjord, ranging from one side of the fjord to to the other, whereas eddies in the outer fjord are much more persistent.

The mean velocity is always directed down fjord and varies from 0.05–0.2 m s$^{-1}$, implying an iceberg residence time in the fjord of 0.5–2 d.

A lack of suitable climate, glaciological, and oceanographic measurements within Johns Hopkins Inlet prevents a detailed analysis of the mechanisms driving the observed flow variability. Nonetheless, our observations are consistent with a general understanding of fjord circulation that has emerged over the past two decades. Subglacial discharge from tidewater glaciers

mixes with warm salty water at depth and drives a buoyancy-driven circulation (Straneo and Cendese, 2015; Truffer and Motyka, 2016). The strength of the buoyancy-driven circulation depends on the magnitude of the subglacial discharge (Carroll et al., 2015), which typically peaks in the afternoon on daily timescales and in mid-to-late summer on seasonal timescales (e.g., Jackson et al., 2022), and the location of the plume may vary seasonally as the subglacial drainage system evolves (e.g., Schild et al., 2016; Cook et al., 2020). Thus, the eddies near the terminus of Johns Hopkins Glacier appear to be controlled primarily

by changes in the subglacial outlet, whereas eddies located in the outer fjord are a result of flow past topographic features. Other processes, such as winds (Straneo et al., 2010) and sill-generated mixing (Hager et al., 2022), can modify the buoyancy-driven circulation pattern so that there is not always a one-to-one correlation between subglacial discharge and fjord or iceberg velocities, especially over longer timescales. Nonetheless, subglacial discharge, through its effect on fjord circulation, appears to be an important driver of variability in iceberg habitat for seals, especially in the near-terminus region and over diurnal

timescales.

Although our limited sample size prevents a more in-depth statistical analysis, aerial photographic surveys of harbor seals that coincided with our time-lapse imagery suggest seasonal differences in the iceberg habitat used by seals. During the pupping season, the seals are rarely found on icebergs exceeding 0.2 m s$^{-1}$. In contrast, during the molting season high concentrations



of seals are found on icebergs exceeding 0.2 m s$^{-1}$. This suggests that mothers may prefer to stay in slow flowing waters that
provide safer and more stable iceberg habitat during the pupping season. However, later in summer during the molting season,
the stability of the ice habitat may be less important as pups have been weaned and the fidelity of seals to the ice habitat is
reduced. Since iceberg velocities and persistence in the fjord are linked to glacier runoff, changes in the timing and duration
of the melt season or intensity of melt or precipitation events may influence harbor seals by reducing the availability of slow
flowing and stable icebergs during the pupping season, which may have implications for young pups that are vulnerable to
predation and still dependent upon their mothers for energy. Furthermore, if ice is moving faster and is less persistent and less
dense, seals may spend more time in the water swimming and repositioning to find more suitable and stable ice which could
result in increased energy expenditure. Spending more time in the water may come at an energetic cost particularly for recently
weaned pups that undergo a post-weaning fast and lose a substantial portion of their body mass and thus are at greater risk of
a negative energy balance due to increased thermal stress (Harding et al., 2005).

Tidewater glacier fjords are dynamic and rapidly changing due to physical processes occurring along the ice-ocean interface
that are driven by local environmental factors in combination with larger-scale climatic forcing. Collectively these physical
changes, which are rapidly occurring, will have downstream impacts and influence nutrient cycling, invertebrate and vertebrate
energetics and life-history, foodwebs, and marine ecosystems in fjords (e.g., Straneo et al., 2019; Hopwood et al., 2020).
Interdisciplinary studies that focus on linking the impacts of physical change to species and biological systems, coupled with
long-term monitoring, will be essential to elucidating how climate change will influence tidewater glacier fjord systems.

*Code and data availability.*   Data are publicly in the Arctic Data Center Repository with the following citations: 1) Jason Amundson. (2022).
Timelapse photos of Johns Hopkins Inlet iceberg habitat, Glacier Bay National Park, Alaska, 2019. Arctic Data Center. doi:10.18739/
A2X921K7T. 2) Jason Amundson. (2023). Timelapse photos of Johns Hopkins Inlet iceberg habitat, Glacier Bay National Park, Alaska,
2021. Arctic Data Center. doi:10.18739/A2VQ2SC1V. 3) Jason Amundson. (2023). Timelapse photos of Johns Hopkins Inlet iceberg habitat,
Glacier Bay National Park, Alaska, 2022. Arctic Data Center. doi:10.18739/A2ZK55N82. The iceberg tracking code, developed by Kienholz
et al. (2019), is available at https://bitbucket.org/ckien/iceberg_tracking/src/master/

*Author contributions.*   JA and JW developed study objectives and secured funding; JA and LK collected the time-lapse imagery; JW and LP
collected, curated and analyzed aerial photographs. AB collected auxiliary and meteorological data. LK led the time-lapse photo analysis
and figure development. JA and LK interpreted the results and prepared the manuscript with contributions from all co-authors.

*Competing interests.*   The contact author has declared that none of the authors has any competing interests.



*Acknowledgements.* This work was supported by North Pacific Research Board award 1905. Glacier Bay National Park and the Southeast Alaska Network provided support for aerial photographic missions. Aerial surveys were carried out under National Marine Fisheries Service permit numbers 358-1787-00, 358-1787-01, 358-1787-02, and 16094-02. Numerous individuals provided field and logistical support including Dennis Lozier, Chuck Schroth, Justin Smith, Christian Kienholz, Nicole Abib, John Harley, Ellie Bretscher, and Caitlyn Montalto.



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
