# Peer review of "Fine-scale variability in iceberg velocity fields and implications for an ice-associated pinniped"

_EGUsphere, 2024_

## Author Response (AR1)

**Reviewer 1**

This is an interesting and well-written study on the biophysics of a tidal glacier fjord habitat for harbor seals in southeast Alaska. The approach is clever as well as an unique and novel application for pinniped habitat studies. Although sample sizes precluded the authors from detailed statistical analysis of links between seals and iceberg velocity within the fjord, there are some interested descriptive analyses and figures presented here.

The bulk of the results focus on describing iceberg movements and velocity within the fjord, which represents considerable effort and resources in itself. That said, it would be helpful though to have a bit more framing on the ecological significance and context of this glacial iceberg habitat for harbor seals in the Introduction. Although the authors did not consider it here, I wonder if there's an ability to estimate iceberg size and shape from any of the imagery used in this study, in terms of iceberg selection by seals and how fjord velocity is related to availability of suitable habitat. I recommend considering these kinds of aspects in the Discussion or possibly adding a section on potential next steps or additional research. There are also 9 figures, and it might be worthwhile to move a couple to supplementary materials and also introduce some quantitative analysis to more precisely link environmental conditions and diel patterns to iceberg velocity. In my line comments, I provide suggestions to improve clarity and details of the study.

**Line comments**

23-31: While this paragraph lays out glacial ice as important habitat for harbor seals in Glacier Bay and that surveys are conducted during pupping and molting season, there could be additional details about the ecological function and purpose that glacial ice plays for harbor seals (i.e. why is it important habitat?). This is unique habitat for harbor seals and so additional introduction to the ecology of this ecosystem would be useful, particularly for readers unfamiliar with this habitat.

> **Yes, we can provide additional context related to the ecological function and purpose that glacial ice plays for harbor seals.**
>
> **Tidewater glaciers in fjords produce icebergs and floating glacier ice that is used by harbor seals. The ice used by harbor seals serves an important ecological function including providing an important substrate for critical life-history functions including pupping, nursing young, molting, resting, and avoiding predators. Tidewater glacier fjords hosts some of the largest seasonal aggregations of harbor seals in the world and provide a refuge from predation. Furthermore, the ice provides a stable platform for caring for and nursing young.**

24 (and later): I deeply appreciate the inclusion and normalization of Tlingit place names here and throughout. Another option might be to put the Tlingit name first in the sentence and then parenthetically the western name, noting hereafter referred to as. For example: "In Sit' Eeti Geeyi (or what is currently known as and hereafter referred to as Glacier Bay),…" Otherwise, please indicate Tlingit in the parentheses for readers who may be unfamiliar with the region, e.g.: Glacier Bay (Sit' Eeti Geeyi in Tlingit).

**We will provide appropriate background for the Tlingit people and adapt our place names to have the Tlingit name to appear first.**

71-74: It's unclear how often (if at all) conditions affected camera imagery, but presumably fog, rain, or other conditions could affect visibility and therefore the ability to detect and track icebergs in images. Please clarify if this is the case and how it was dealt with if so.

**In general, fog and rain did not significantly impact our timelapse image analysis. However, on a few occasions our method was affected by particularly heavy fog or wind-driven rain that obscured the camera lens. This occurred only 6 times across our two field seasons of data collection. We will include these details in the methods section to clarify the limitations of our approach.**

90-91: Figure 2b seems to contradict the text here for flights in 2022 (although this line is referring to 2019), with several image footprints that appear to overlap, presumably where the aircraft is turning near the glacier terminus. Please clarify in the next paragraph about 2022 flights if the differences in survey methods between years introduced the potential for overlapping images and therefore double or multi-counting of seals.

**Images were visually inspected by a trained observer to account for and remove any overlap in 2022 and to ensure that double-counting didn't occur. See details on Womble et al. 2020; 2021. There was no overlap in imagery from 2019. We will clarify this in the manuscript**

104: please clarify what kind of shapefiles were created. Was it points with a single location for seals? Or if it was polygons or something else, provide more detail about how these were created.

105-106: In other words…two sets of shapefiles, specifically for pups as well as non-pups, were created during pupping season in both years of aerial surveys? Only non-pup shapefiles, in each year of surveys, were created during the molting season?

**Shapefiles were created using ESRI ArcGIS. Each seal in each image was mapped resulting in a latitude and longitude position for each seal. See detailed methods described in Womble et al. 2020; Womble et al. 2021. Pups and nonpups were identified during the pupping season in June. All seals were identified as nonpups during the molting season in August, because pups have increased rapidly in size and can no longer be distinguished from nonpups. We will add specifics about this into the final version of the manuscript**

Please also include parenthetically which months or time period were considered pupping and molting seasons – was this the same period in 2019 and 2022?

**We mention the 2019 molting and 2022 pupping seasons earlier in line 25 and lines 114-115 and in 164-169.**

143 and Fig 3: This is such an interesting figure, and tells so much information. However, I'm a little confused about how the "prominent, persistent" eddies were identified. The text suggests that the placement of the eddies outlined in the boxes were persistent across days/time, but I don't see each of

these 3 locations with the eddies for each day in Fig 3. So maybe I'm missing across what time scale an eddy was considered to be persistent.

**Each streamline plot is the daily average for a particular date and eddies that show up on this timescale have to be persistent for the majority of the day. The three eddies outlined in Fig. 3 are eddies that are persistent across a particular day and frequently found on various days throughout the time series. We will clarify this in both the caption and text.**

153: The order of figure numbering is hard to follow – it doesn't seem like figures are presented in the order in which they are referenced in the text. In this case, the text jumps from Fig 5 to Fig 8.

**See our following comment about paper reorganization.**

Figs. 4 and 7: there are a lot of figures in this paper, and I wonder about some that could be moved to supplementary material because they are not primary contributors to the key points. These figures may be good options for the supplement. These figures are also challenging to line up the environmental data with the velocity data, and I wonder about providing statistics to support the statements about correlations between environmental data and velocity, which can then be used to reinforce the visual interpretation as well as may be able to be summarized more succinctly in a table for example.

**We believe that Figs. 4 and 7 are important because they show a correlation between climate and fjord currents, but only on short time scales. To make this point more salient and address the above comment about reference numbers we intend to expand on the discussion of the figures and reorganize the results section . The reorganization will be something as follows: (i) iceberg velocities pick up during the day due to runoff and katabatic winds (Fig. 7), which (ii) manifests itself as changes in circulation patterns (Fig. 6). However, (iii) the pattern doesn't hold for longer time scales (Fig. 4), and therefore we can't simply assume that summers with more runoff will result in higher iceberg evacuation rates.**

179-190: The use of CCDFs is an interesting approach for describing seal iceberg velocity presence, given the small sample size in the number of days sampled. Yet Fig 8 provides a nice visual representation showing seals affiliation with plumes, and it raises the question of whether iceberg velocity is the correct predictor of seals in the fjord and hauled out on icebergs. Another potential interesting predictor, not considered here, would be proximity to plumes.

**This is an excellent observation, and exploring a correlation between seal proximity and plume location as an additional predictor would be insightful. However, our current methodology for identifying plume boundaries—visually delineating them from Planet imagery—has limitations. Defining the edge of the plume is inherently subjective, as some areas appear highly turbid and visible in satellite imagery, while others are diffuse and harder to discern. This makes assessing seal proximity to the plume's edges problematic. A better approach would involve deriving spatial turbidity using remote sensing techniques, such as the method described in Tavora et al. (2023). However, this method is complex and beyond the scope of our current study. We will, however, incorporate the idea of**

**investigating correlations between seal locations and turbidity values into our discussion, particularly in the context of potential future research directions.**

Fig. 8: during the pupping period, can pups on ice be distinguished from the adults on ice? It's mentioned at line 105 that there were different shapefiles for pups and non-pups, so if possible, it would be interesting to describe if there are differences in use of icebergs, and relative to the plume for pups (or mom-pup pairs) and non-pups.

**Yes, during the pupping period, pups and nonpups can be distinguished. However, during the pupping period, pups are predominantly hauled out on the ice with their mothers (nonpups) as mother-pup pairs, so we typically do not see differences in the use of icebergs or plumes by pups and nonpups during June. So, we wouldn't expect to see differences in use of icebergs since pups and their mothers are typically located together as pups are still dependent upon their mothers for nutrition. Occasionally we observe lone pups, but it is much less common. Lone pups may be observed if the mother is away foraging or if the pup has been recently weaned.**

Fig 9: Please include a description of what the y-axis, $P(V>v)$, means in the caption, including V and v.

**The y-axis indicates the likelihood that speed V of a randomly selected seal will have a speed that is greater than v, the values along the x-axis. We will clarify this in our forthcoming revisions.**

286: Are the aerial survey seal data also publicly available? Please include references for users interested in seal data.

**Shapefiles from aerial surveys are archived at the National Park Service and can be accessed by request by contacting Jamie Womble. Details will be added to the data availability section at the end of the manuscript.**

**Reviewer 2**

**General Comments:**

This paper explores pinniped distribution in an iceberg habitat by examining the relationship between glacial fjord velocity and harbor seal distribution. Key takeaways include how seal distribution is influenced by glacial velocity and outflow plums; seals are more likely to be found on the edges of plumes, in relatively slower moving water during the pupping season, and relatively faster moving water during the molting season. It is also discussed how adult harbor seals are not likely to be energetically impacted by swimming against glacier currents, but recently weaned pups may be negatively impacted if they have to spend more time in the water due to changing glacial habitat conditions. This paper gives important and novel insight into how changes in the glacial ecosystem due to climate change (i.e. increased precipitation and glacial calving) may impact harbor seal distribution and behavior. This information improves our understanding of how ice-associated marine mammal species may be influenced by ongoing ecological shifts, which is vital to inform future conservation efforts.

I believe there are aspects not discussed that would enhance the paper, particularly a more detailed exploration of potential future research directions. Should more effort be made to survey and conduct this research for the month of July? Since the paper mentions that the mean velocity in July is greater compared to June and August, would this give valuable insights into how glacial surface currents impact distribution? What future statistical analyses could be done to add to these results?

Furthermore, given that other studies have found that the haulout behavior of ice-associated pinnipeds is impacted by environmental variables (ex. Hamilton et al., 2014 published in Plos One), I also think there needs to be more of a discussion about potential covariates that may impact seal distribution rather than solely glacial fjord velocity and plumage area, especially since seals are more likely to be seen while hauled out vs. when swimming (Line 172). Glacial fjord velocity and plumage area are only a part of the bigger picture. Lastly, there is no discussion of why seals may select for recently calved icebergs (Line 175 & Figure 8); is it because they are larger, more stable, etc.?

> **The reviewers are correct that there are a range of covariates that influence the haulout behavior of harbor seals including time of day, day of year, season, weather, etc. These covariates have been well-documented in others studies in tidewater glacier fjords (e.g., Mathews & Pendleton 2006; Blundell & Pendleton 2015; Mathews et al. 2016), so we did not focus on those specific covariates. Instead, our efforts were aimed at the influence of ice and physical variables within the fjord, including plumes and velocity fields, that may influence ice movement. Furthermore, in addition to covariates mentioned above there may be additional social factors that may influence the haulout behavior of seals using ice as well as anthropogenic factors including vessels which may disturb seals in glacial ice habitats (e.g., Jansen et al. 2015).**

**Specific Comments:**

Line 19: Clarify "given the dramatic changes in ice coverage"; is this referring to the overall reduction in circumpolar ice concentrations? Seasonality shifts?

> **By "dramatic changes in ice coverage" we mean the overall reduction of ice coverage in the circumpolar regions. We will clarify this in our edits**

Line 24: Add "...(Sít' Eetí Geeyi in the Tlingit language)" ; no mention of the Tlingit people is given throughout the paper

> **This is addressed in a response to Reviewer 1 and we will add context about the Tlingit people to the paper.**

Line 56: How much overlap between photos?

> **There was no overlap between photos in 2019 as the survey was designed to ensure there was not overlap. Any overlap in 2022 was removed before counting to seals to ensure that seals were not double-counted.**

Line 63: What does an initial guess mean in this context?

> **We mean an initial guess of each of the four free parameters used for the camera initialization. We will clarify this in our edits.**

Line 105-106: Provide clarification; does this mean pups were not included in the data processing for the molting season? Or all seals in imagery were processed as non-pups?

**Only nonpups are counted during the molting season as pups have increased in size and mass and can no longer be effectively distinguished from nonpups. During June, seals are classified as pups and nonpups; however, during August all seals are classified as nonpups (see details in Womble et al. 2020; 2021).**

Figure 3: Why were these days selected to show the velocity fields? Should the aerial survey day velocity streamlines also be included in this figure?

**We chose those days because we felt that they best characterized the range of velocity fields that we observed across both field seasons. We will clarify this in our caption. We didn't include the aerial survey dates here because they weren't the best examples of daily averages. We do however show this later in Fig. 8.**

Figure 8: Define time frame for "recently" calved icebergs.

**We consider recently calved icebergs to be those near the terminus that have not yet dispersed throughout the fjord and likely calved within a few hours of the satellite image being taken. We will add a better description of this in the Figure caption.**

Line 216: It was not clear that 2019 was used to represent the pupping year and 2022 to represent the molting year due to small sampling size until I read the caption under Figure 9. Potentially make that more clear here, earlier in the paper.

**We mention the 2019 molting and 2022 pupping seasons earlier in line 25 and lines 114-115 and in 164-169.**

Lines 211-214: Seals on slower moving icebergs during the pupping season does not explain why males are on slower moving ice; thoughts on this would add to the discussion.

**During the pupping season in June, the majority of seals observed in fjords are mothers and pups. However, males may begin to move into the fjord for breeding after the pups are weaned.**

Lines 217- 243: Much of the discussion focuses on the potential energetic demands of glacial surface currents on harbor seals, but this is the first time energetic demands are mentioned. Incorporating this idea earlier, perhaps in the introduction, would enhance the paper's overall cohesiveness.

**We agree that incorporating additional background on energetic demands into the introduction will enhance the cohesiveness of the paper. We will revise the introduction accordingly.**

**Technical Corrections:**

Line 85: Repetitive with above; potentially cut

**This was fixed in the first round of edits sent to the editor**

Line 145: Place "Each column in Figure 5…" sentence in figure 5 caption

**We will incorporate this advice into the forthcoming revision.**

Line 204-207: Move this paragraph to the first paragraph of discussion (line 192-197) to improve flow

**This is a great suggestion and we will incorporate this advice into the forthcoming revision.**

Line 219: "Between" is written twice

**This was fixed in the first round of edits sent to the editor**

Figures are not always in the order they appear in the text. For example, Fig 8 is mentioned before Fig 7

**We discuss this issue in an earlier comment. We will address this during our reorganization of the results section.**

**Editor Report:**

Referee #1 wondered about the ability to estimate the seal's most preferred iceberg size and shape, perhaps as a next step of future research. Please indicate what was your response to this.

**Previous studies have investigated relationships between seals and iceberg size and found no correlation (Womble et al. 2021, Frontiers in Marine Science). However, future work might consider addressing how iceberg shape may influence use of icebergs by seals. Opportunistic observations suggest that seals tend to use icebergs that are flatter with less topographic relief which allows for easier access when transitioning from the water to the ice. This line of inquiry, while interesting, is not directly related to the current study's objectives, and therefore, we do not feel that proposing it as a next step for future research is necessary. We will expand on other future research steps more inline with the focus of this paper.**

Referee #2 suggested elaborating on "potential future research directions", such as more survey effort in "the month of July" and associated "statistical analyses". Please clarify what was your response to this. Should more effort be made to survey and conduct this research for the month of July? Since the paper mentions that the mean velocity in July is greater compared to June and August, would this give valuable insights into how glacial surface currents impact distribution? What future statistical analyses could be done to add to these results?

**Aerial photographic surveys of seals were conducted during the pupping season in June and during the molting season in August. This is a widely accepted approach for monitoring the distribution and abundance of harbor seals in Alaska (e.g. Boveng et al. 2003, Marine Mammal Science) as a higher proportion of seals tend to be hauled out or ashore during June and August for critical life-history events (pupping, molting). There have been some counts conducted during July in Johns Hopkins Inlet during 2007 and**

**2008, which suggests that the number of seals decreases in July, after pups are weaned (Young et al. 2014, Tourism in Marine Environment). We agree that greater survey coverage in July would be interesting, given the increase in surface currents. We will elaborate on future statistical analyses that would be beneficial. In particular, our study was limited to aerial and time-lapse observations; incorporating oceanographic measurements in future research and focusing on a statistical analysis of seal distribution and their correlation to variables such as water temperature and salinity would provide would be a good next step.**

Referee #2 did not seem to ask you to focus on well-documented covariates but instead suggested acknowledging such potential confounding environmental variables in discussion to show that your focus corresponds only to "a part of the bigger picture." So, I could not follow what changes in the manuscript you are planning to make in this regard, adding more discussion of other environmental and social factors or omitting them?

**In addition to velocity fields, other environmental covariates including time of day, weather, and tide may influence the distribution and abundance of seals and have been previously described in Mathews & Pendleton 2006, Marine Mammal Science; Boveng et al. 2003, Marine Mammal Science; Blundell & Pendleton 2015, PlosOne). In addition to physcal factors related to ice and other environmental covariates, there are likely social factors related to mother-pup behavior that may influence the distribution of seals. Collectively, the distribution of seals in the fjord likely reflects a complex combination of abiotic (ice, tides, velocity fields, weather) and biotic factors (age, sex, presence of pup, group size etc).**

Finally, I am unsure what your response was to this question "why seals may select for recently calved icebergs."

**Seals may prefer these icebergs due to the higher concentration of ice in the area following a calving event, providing them with a greater number of icebergs to choose from as an ideal habitat.**

**Citations**

Tavora, Juliana, et al. "Detecting turbid plumes from satellite remote sensing: State-of-art thresholds and the novel PLUMES algorithm."*Frontiers in Marine Science* 10 (2023): 1215327.